# Factors That Contribute to hIAPP Amyloidosis in Type 2 Diabetes Mellitus

**DOI:** 10.3390/life12040583

**Published:** 2022-04-14

**Authors:** Adriana Sevcuka, Kenneth White, Cassandra Terry

**Affiliations:** Molecular Systems for Health Research Group, School of Human Sciences, London Metropolitan University, London N7 8DB, UK; ads0520@my.londonmet.ac.uk (A.S.); kenneth.white@londonmet.ac.uk (K.W.)

**Keywords:** human islet amyloid polypeptide, amyloidosis, type 2 diabetes mellitus, aggregation

## Abstract

Cases of Type 2 Diabetes Mellitus (T2DM) are increasing at an alarming rate due to the rise in obesity, sedentary lifestyles, glucose-rich diets and other factors. Numerous studies have increasingly illustrated the pivotal role that human islet amyloid polypeptide (hIAPP) plays in the pathology of T2DM through damage and subsequent loss of pancreatic β-cell mass. HIAPP can misfold and form amyloid fibrils which are preceded by pre-fibrillar oligomers and monomers, all of which have been linked, to a certain extent, to β-cell cytotoxicity through a range of proposed mechanisms. This review provides an up-to-date summary of recent progress in the field, highlighting factors that contribute to hIAPP misfolding and aggregation such as hIAPP protein concentration, cell stress, molecular chaperones, the immune system response and cross-seeding with other amyloidogenic proteins. Understanding the structure of hIAPP and how these factors affect amyloid formation will help us better understand how hIAPP misfolds and aggregates and, importantly, help identify potential therapeutic targets for inhibiting amyloidosis so alternate and more effective treatments for T2DM can be developed.

## 1. Introduction

Diabetes mellitus is a complex condition linked to increased blood glucose levels, compromised insulin production or action, that can be subdivided into five major types: monogenic diabetes, Type 1 Diabetes mellitus (T1DM), Type 2 Diabetes Mellitus (T2DM), gestational diabetes and, more recently, Type 3 Diabetes (T3DM) [1]. Monogenic diabetes is rare and arises from mutations in one of about a dozen genes, including the insulin gene and genes downstream of insulin action, such as glucokinase [2]. T1DM is classified as an autoimmune disorder in which T-cells attack and destroy pancreatic β-cells, which are the site of insulin secretion [3]. Gestational diabetes can occur during pregnancy when the natural production of diabetogenic hormones from the placenta, which induces insulin resistance, cannot be compensated sufficiently by increased insulin production from the mother [4]. In addition to the well categorised conditions T1DM and T2DM, some scientists have recently suggested another type of diabetes called ‘Type 3 Diabetes’. T3DM has been used to describe the neurodegenerative pathologies observed in those with Alzheimer’s Disease, accelerated by insulin resistance in brain tissue, thought to be due to T2DM [5].

T2DM is a complex metabolic condition characterised by hyperglycaemia and peripheral insulin resistance, leading to depletion and dysfunction of pancreatic islet β-cells [6]. A range of co-morbidities has been linked to T2DM including cardiovascular disease, sensory neuropathy, stroke, kidney failure and blindness [7]. The World Health Organization conducted a worldwide study on diabetes in 2016 [8]. The report showed that the number of adults currently living with diabetes is over 422 million worldwide and this has quadrupled since 1980. T2DM accounts for 95% of all these cases, resulting in over 1.5 million deaths. This increase has been linked to rises in obesity that has resulted in a 5% increase in mortality rate in those under the age of 70, with complications linked to blindness, stroke, heart attack, kidney failure and lower limb amputations.

T2DM has been associated with several risk factors that are genetic, epigenetic and environmental [9]. More than 140 genome risk variants have been identified through meta-analysis of genome-wide association studies (GWAS) in European populations [10] and a further 318 novel variants associated with T2DM were found in a trans-ethnic GWAS [11]. Environmental and lifestyle factors have been commonly linked to the rapid increase in T2DM cases worldwide. Obesity has been singled out as one of the most significant factors contributing to the development of T2DM. Almost one-third of the global population can be classified as obese, seeing a meteoric rise in the last century [12]. The term “diabesity” encompasses the co-existence and link between T2DM and obesity, highlighting the fact that being obese is one of the biggest risk factors for T2DM [13]. T2DM was previously assumed to be an adult-onset condition; however, the rise in childhood obesity (which is linked to increasingly sedentary lifestyle and glucose-rich diets) has seen cases of T2DM in under 18-year-olds rising [14]. Diet is therefore also another important contributory factor, with low-fibre, high-glucose diets being associated with a higher risk of T2DM [15].

Social-economic factors are also a risk factor linked to T2DM. Individuals from low-socioeconomic communities are more likely to suffer from chronic stress which is associated with increased blood pressure and elevated blood glucose levels [16]. Those from low-socioeconomic backgrounds are also more likely to have poor diets and be obese, which also accounts for the rise in T2DM in these communities.

Microorganisms have also been associated with T2DM, as highlighted in recent studies looking at whether the gut microbiome is linked to T2DM [17]. Elevated levels of bacteria such as *Fusobacterium* and *Ruminococcus* have been detected in T2DM patients compared to non-diabetic patients. These bacteria form part of the microflora of the intestines that are thought to regulate levels of lipopolysaccharides (LPS), which have been linked to the advancement and development of T2DM [18]. Levels of circulating LPS are increased in T2DM patients and the liver of LPS induced mice has been observed to show insulin resistance. Additionally, healthy weight mice showed fasting glucose levels and weight gain increase in the same range as those of obese and diabetic mice [19].

Prescribed medications to manage T2DM include antihyperglycemic drugs such as metformin and sulfonylureas. Although these have proven to reduce mortality and assist in the control of blood glucose levels, metformin has associated side effects such as nausea and diarrhoea in 50% of patients. This is thought to be as a result of metformin increasing levels of glucagon-like peptide 1 (GLP-1) and increasing intestinal glucose turnover, as well as influencing the microflora of the gut [20]. Sulfonylureas promote insulin release independently of glucose levels but have been reported to cause hypoglycaemia [21]. Scientists are therefore currently researching more effective treatments that focus on reducing and/or reversing the damage to β-cells caused by T2DM and ways to reduce any associated side effects. The development of preventative treatments for pre-diabetic patients would be more effective as it would prevent T2DM and the need for medication and would prevent complications affecting other parts of the body (e.g., eyes, heart) associated with this complex disease.

In order to progress with therapeutic interventions and help prevent disease onset, we urgently need a better understanding of T2DM at the molecular level. Despite a plethora of scientific literature on diabetes, exactly what initiates the disease and what factors are responsible for accelerating the condition are still unclear. In this review, we have focused on the role of hIAPP and its tendency to misfold and aggregate in the molecular pathology of T2DM, since accumulating evidence suggests that different conformations of the protein are associated with the initiation and/or acceleration of the disease.

## 2. Main Text

### 2.1. The Human Islet Amyloid Polypeptide and Its Link to Diabetes

Deposits of incorrectly folded (“misfolded”) hIAPP in the pancreas of people with T2DM have been reported in up to 90% of all T2DM patients [22]. The change in protein conformation from a soluble protein to misfolded forms, which can result in amyloid fibrils in the process of amyloidosis, is associated with several diseases such as Alzheimer’s Disease and T2DM, where they aggregate and damage cells and surrounding tissues. Misfolded forms of hIAPP have been linked to pancreatic β-cell damage and as a result, decreased release of insulin and impaired glucose regulation, which can lead to T2DM [23].

hIAPP is a 37 amino acid protein with an element of intrinsically disordered structure and is secreted from β-cells together with insulin [24]. It is classified as an intrinsically disordered protein (IDP) that lacks stable secondary or tertiary structures under physiological conditions, since it contains intrinsically disordered regions within its structure [25].

The gene that encodes hIAPP is found on chromosome 12, includes 3 exons [26] and is initially expressed as an 89 amino acid residue long pre-pro-peptide [27]. A 22-residue signal sequence is cleaved, forming a 67-residue propeptide, which is then processed in the Golgi, followed by the secretory granules of β-cells [28]. Nine amino acids are cleaved from the N-terminus and 16 from the C-terminus of the protein by prohormone convertase enzymes. The remaining 5 mostly basic amino acids are removed by carboxypeptidase E to produce the secreted peptide [28,29].

Several functions have been ascribed to IAPP, including suppression of glucagon release, control of gastric emptying and regulation of satiety [30], all of which can influence glucose homeostasis. IAPP has also been suggested to help regulate blood glucose levels through inhibitory effects on insulin secretion, although there is some ambiguity in the evidence supporting such a role [30,31,32]. Interestingly, a study in mice in which the IAPP gene had been ablated suggested a protective role for IAPP in β-cell function [33]. IAPP has pleiotropic effects and is able to cross the blood–brain barrier [28,29], and hence there is potential for aggregation events occurring around the body that may contribute to the development of diabetes.

### 2.2. The Structure of hIAPP

Understanding the structure of hIAPP and its various conformations is important for understanding how it contributes to T2DM and hence for designing effective therapeutics. In T2DM, evidence suggests that hIAPP can adopt different conformations, including pre-IAPP (an un-cleaved form), monomers, dimers, oligomers and fibrils [34]. Like other protein misfolding disorders (PMDs), it is unclear to what extent each protein conformation contributes to toxicity and disease.

In vitro, synthetic hIAPP monomers dissolved in phosphate buffer or diluted hexafluoro-2-isopropanol form random coil conformations, which misfold into β-sheet structures and fibrils [25]. Analysis of hIAPP fibrils using site-directed spin labelling and electron paramagnetic resonance (EPR) spectroscopy, in combination with simulated annealing molecular dynamics, revealed the fibrils to be composed of stacked misfolded forms of hIAPP. Each hIAPP was bent into a hairpin with each arm having a region of β-sheet structure, opposite and staggered against each other. The fibrils had a left-handed twist and a hydrophobic surface at the fibril ends that is suggested to be the site of fibril growth where subsequent hIAPP monomers can bind, leading to fibril elongation [35]. Molecular dynamics simulations confirmed the random coil overall structure of hIAPP and revealed many metastable conformational states [36]. Some of the states had β-hairpin secondary structure and hydrophobic surfaces, features which facilitate nucleation of aggregation pathways. The formation of insoluble hIAPP fibrils from soluble monomers is proposed to occur via nucleated conformational conversion or conformational selection [36]. Nucleated conformational conversion involves the assembly of fibrils from oligomeric intermediates such as protofibrils [37]. Conformational selection means that the binding of monomers does not involve any change in conformation state but binds to an already formed conformation [38]. NMR results suggest that hIAPP undergoes nucleated conformational selection where the N-terminal α-helical region is involved in monomer “collapse” (hydrophobic or hydrophilic collapse), suggesting that monomers first collapse before further rearranging into ordered proto-fibrils composed of β-sheets [39]. Conformational selection involving this selective collapsing of monomers possessing β-sheet structures has also been supported by molecular dynamic studies [37].

Recently, several cryo-EM structures of recombinant hIAPP have been produced [40,41,42,43] that show fibrils are composed of two protofilaments. The structures observed suggest that the process of fibrillization may be synchronous [43] and the structural similarity of hIAPP fibrils to polymorphs of amyloid-β (Aβ) fibrils [42] may explain how both fibrils can cross-seed one another.

The most detailed, and biologically relevant structures of hIAPP fibrils was published recently by Cao et al., 2021 [44], showing eight morphologically different cryo-EM fibril structures seeded from fibrils extracted from islet cells from a T2DM donor. Four of the structures revealed twisted fibrils comprising two linked protofilament chains each composed of stacked β-strands. Two of the twisted fibrils are homodimers, whilst the other two are heterodimers. Two core folds can be observed in the fibrils which were observed to be capable of interacting in three different ways to form protofibrils, thus allowing the different structures to form. Some of the seeded fibrils do not resemble any of the unseeded ones (derived from peptides only), hence, they could resemble the structure of hIAPP in vivo [44].

Since membranes have been suggested as crucial to hIAPP misfolding and aggregation [45], it is essential to characterise the structure of hIAPP when bound to a membrane-like environment. Studies suggest that membranes accelerate the transition from α-helical to β-sheet pleated structures as revealed by NMR, EPR, TEM and Atomic force microscopy (AFM) studies [46,47,48,49]. NMR and EPR experiments using hIAPP bound to micelles highlighted residues 9–22 as having an unstable α-helical region, which is assumed to become stabilised during the aggregation process [34]. Lipid binding assays and TEM showed that when bound to anionic or zwitterionic membranes, hIAPP transitions from an α-helical structure to a β-sheet structure within minutes [50]. AFM has also been used to observe membrane-bound pre-fibrillar hIAPP, showing structures resembling ion-channels which could damage membranes by forming pores and changing their fluidity [51]. Other AFM studies have shown hIAPP accumulating on membranes in the form of unstructured aggregates. Membranes containing cholesterol revealed less bound amyloid compared to cholesterol-free membranes [52,53], highlighting the importance of membrane composition to amyloidosis.

Importantly, decreased insulin degrading enzyme (IDE) levels can lead to hyperinsulinemia, which may contribute to insulin resistance [54]. IAPP is a substrate for the insulin degrading enzyme (IDE) [55], a zinc metalloproteinase expressed in most cell types and also found in the circulation. Modulation of IDE activity could have consequences for amyloid formation by hIAPP and other amyloidogenic proteins [56]. Inhibition of IDE increased amyloidosis and hIAPP cytotoxicity in a cell culture model [57], but not in cultured islets [58]. In mice models, acute administration of a potent IDE inhibitor demonstrated beneficial effects on glucose tolerance [59], indicating potential for therapy of T2DM. However, given its wide distribution and other biochemical functions such as a chaperone and ubiquitin-proteosome pathway modulator, long term or untargeted use of inhibitors may be problematic [57,60].

### 2.3. Important Residues Involved in hIAPP Aggregation

Different mechanisms have been proposed explaining how monomers stack together and assemble into oligomers. Identifying exactly how the different conformations form and the key residues involved is important for developing therapeutics to inhibit aggregate formation, especially at the early stages of the process before extensive cellular damage occurs. NMR experiments using hIAPP peptides in solution suggest the initial events may involve adjacent monomers binding via noncovalent interactions involving residues 26–32 before assembling into oligomers [61]. Another study suggests a similar region, residues 29–33, as key to oligomer assembly, and also highlights residues 20–27 as important in fibril formation [62,63].

Several studies indicate that the amyloidogenic properties of human IAPP are linked to residues 20–29, one of the regions previously identified as key to oligomer assembly [64]. Residues 8–18 have also been identified as important in determining the side chain orientations along the β-strand within fibrils that have been shown to differ in fibril polymorphs produced in vitro [28,65]. A comparison of the primary sequence of human IAPP to that in rats (that do not form amyloid or get T2DM) reveals important differences (Figure 1). There are no proline residues in hIAPP, whereas rat IAPP has prolines at residues 25, 28 and 29. Molecular dynamics simulations demonstrated that proline residues show a low propensity of forming of β-sheet structures [64] and explains why hIAPP can more easily form β-sheet structures compared to rats. This provides compelling evidence associating IAPP aggregation with development of diabetes. This has been further corroborated in studies with transgenic rodent models, where rats that do not normally develop diabetes can become naturally diabetic if the human IAPP gene is inserted [66]. We compared the tertiary structures of human (PDB 5MGQ) and rat IAPP (PDB 2KJ7) solved by NMR and highlighted where in these 3D structures residues differed. Residues 18, 23, 25, 26, 28 and 29 differ between the two species.

A Ser20Gly mutation found in Chinese and Japanese populations has been linked to an increased rate of amyloid aggregation, and an increased propensity to T2DM, further supporting the importance of the 20–29 region in amyloidosis [67]. The quantity of amylin produced is unaffected by the mutation, since mRNA levels observed in individuals with and without the mutation were comparable [68]. Cryo-EM structures reveal differences between wild type and S20G fibrils [40,41,42], both showing two protofilaments and a left-handed twist but differences in features such as repeat and crossover lengths. Differences observed in the backbone conformations could explain how the early onset T2D IAPP genetic polymorphism S20G can aggregate more readily than wildtype and may provide a structural explanation for surface-templated fibril assembly [41].

A conserved disulphide bond found outside the cross β-core is located between residues Cys 2 and Cys 7 of hIAPP. Removal of this bond through mutation or Cys-Cys disulphide reduction was shown to increase hIAPP amyloidosis while also decreasing hIAPP-induced membrane leakage, highlighting the importance of this region in fibril formation and related cytotoxicity [69].

### 2.4. The Role of hIAPP Oligomeric Intermediates in T2DM

Accumulating evidence suggests that hIAPP oligomers play a crucial role in cytotoxicity. Due to their soluble nature and short lifespan, it has been extremely difficult to isolate them to characterise their structure. Altamirano-Bustamante et al. [70] used various techniques (Immunoblotting, TEM, ultrastructural immunolocalization and CD spectroscopy) to confirm the presence of oligomers in sera of healthy, T1DM, T2DM and obese children, utilising specific anti-oligomer antibodies. Samples from all groups showed the presence of fibrils and trimers, hexamers and dodecamers and oligomers of varying size were also observed (detected with anti-oligomer antibodies). Surprisingly, oligomers were also detected in sera from healthy children, but they had the lowest number of small oligomers out of all the groups, which are thought to be more toxic than the larger counterparts. The fact that oligomers have also been seen in healthy individuals is confusing, demonstrating that they are present in asymptomatic individuals and may not always contribute to disease. The observation of hIAPP oligomers in sera now requires further experiments to optimise purification methods to try to isolate them from sera so that they can be further characterised.

### 2.5. Mechanisms of hIAPP Cytotoxicity

HIAPP has been linked to cytotoxicity of pancreatic β-cells, but more recently other cells such as neuronal cells [71]. Whilst several mechanisms linking hIAPP and toxicity have been suggested (as summarised in Figure 2), the forms of hIAPP that mediate this toxicity is still debated, as are the precise mechanisms involved. Some studies propose that it is the fibrils that directly damage β-cells [72], while others suggest oligomeric intermediates are responsible [67]. However, more than one form of hIAPP may contribute to cytotoxicity, possibly to different extents that may be via a variety of mechanisms. Monomeric hIAPP has been observed to increase membrane fluidity, leading to membrane destabilisation [73], and has also been observed to increase production of reactive oxygen species (ROS) in β-cells [74]. Oligomeric hIAPP has been shown to decrease cell viability as well as increase membrane fluidity. Cell damage involving fibril growth at the membrane is linked to membrane leakage [75].

### 2.6. Acceleration of hIAPP Misfolding and Aggregation

A crucial question in the understanding of hIAPP aggregate formation is what initiates and accelerates the formation of the fibrils that have been linked to T2DM. If we knew what triggered the misfolding cascade in the first place, then T2DM (and possibly other PMDs) may be preventable. Distinguishing between what initiates and what accelerates the process of amyloidosis is key but may involve the same factors. Several factors may contribute to the formation of these misfolded aggregates, and we have summarised these in Figure 3. These include the concentration of hIAPP, cell stress, the role of proteoglycans, the immune response and the association with other amyloidogenic proteins such as cross-seeding (Figure 3). We have summarised the key findings behind each factor below.

#### 2.6.1. Concentration of hIAPP

There is an important relationship between the concentration of hIAPP and the rate of aggregation [76]. When a threshold concentration is exceeded, there is a huge increase in the rate of aggregate formation. This may be due to nucleation-dependent polymerisation. The critical concentration of hIAPP to initiate aggregation in vitro is approximately 2 μM, which is comparable to circulating plasma hIAPP concentrations recorded in the range 1 to 10 pM [77] and mM concentrations in secretory granules [78]. The high concentration in granules suggests there are mechanisms in place within the granule to prevent aggregation under normal proteostasis [32]. However, when there is a state of insulin resistance the β-cells respond by increasing the release of insulin and hIAPP. The enhanced synthesis of secretory granules required to maintain higher levels of insulin may induce cell stress and induce aggregation of hIAPP during granule assembly, which is discussed further below. Additionally, the abnormal enhanced secretion of hIAPP could result in local concentrations after release high enough to reach the critical concentration necessary for the formation of oligomers [77].

#### 2.6.2. Cell Stress

A proposed accelerator of hIAPP aggregation is cell stress. Endoplasmic reticulum (ER) stress is a known factor in T2DM pathology [79]. An ER stress response is generated when free fatty acid (FFA) levels increase, as observed in obese individuals [80]. In T2DM individuals, the ER cell environment is highly reduced, which in turn reduces the Cys-Cys disulphide bridge of hIAPP proteins. Reduced hIAPP is more prone to forming fibrils as it lacks the propensity to form an α-helix at the N-terminus such as in the oxidised form, as shown by NMR and CD reported by Camargo et al. [81]. The reduction in hIAPP during cell stress is thought to combat ER stress by acting as a buffer. However, failure to remove the reduced hIAPP by lysosomal autophagy can result in a build-up of the reduced peptide, which has a higher propensity to aggregate, leading to a build-up of aggregates and β cell death [81]. There are limited studies outlining the formation and structure of secretory granules; however, the packing of hIAPP in secretory granules may influence hIAPP aggregation. Defective packaging of hIAPP within secretory granules may occur as a result of β cell stress and this defective packaging may assist hIAPP in forming aggregates. More studies are needed, however, to investigate whether this is the case.

Another form of cell stress found in T2DM is oxidative stress that can be caused by chronic hyperglycaemia. As evidenced by methods such as mass spectrometry and high-performance liquid chromatography, markers of oxidative tissue damage have been identified in plasma taken from individuals with T2DM [82,83]. Oxidative stress has been linked to an increase in β cell ROS [84]. As glucose is the leading energy source for the Electron Transport Chain (ETC), hyperglycaemia found in T2DM patients increases the amount of available glucose for the ETC, leading to higher levels of ROS. The resultant increase in ER stress causes misfolding and accumulation of hIAPP polypeptides [85]. A decrease in oxidative stress was observed when inhibiting amyloid aggregation, suggesting amyloidosis may further contribute to even higher levels of oxidative stress [86]. An increase in ROS due to hyperglycaemia has also been linked to vascular-related morbidity in T2DM patients through damage to endothelial cells [87,88].

#### 2.6.3. The Role of Chaperones

Chaperones are proteins required for the correct folding of other proteins and, thus, play an important role in preventing protein misfolding and aggregation. Molecular chaperones are important in response to ER stress [89] in which they are upregulated, and defective expression has been linked to the accumulation of aggregates [90]. The degree of chaperone upregulation in response to stress reduces with age, which is an important factor associated with chronic conditions such as T2DM [50,91]. Reduced levels of ERp55 and ERp57 chaperones that contribute to preventing free oxygen radical-induced damage, have been observed in aged male rats [92]. Treatment of rat pancreatic β-cells expressing hIAPP with high levels of glucose showed an increase in ER stress. Following treatment with a glucose-regulated molecular chaperone protein and a chemical chaperone (taurine-conjugated ursodeoxycholic acid), β-cells showed an increase in insulin secretion and depletion in ER stress [93]. Chaperones are able to form bonds with unfolded forms of proteins to prevent the proteins from interacting with different compounds. Molecular docking studies by Fernández-Gómez et al. [94], have identified chaperones binding to residues 11–28 of hIAPP through Van der Waals forces, hydrophobic interactions and hydrogen bonds. These interactions are suggested to prevent hIAPP dimer formation. Pharmaco-chaperones are able to accelerate or inhibit aggregation, suggesting a potential therapeutic target. Molecules containing aromatic ring structures have been observed to disrupt amyloid structures through the formation of aromatic interactions [94]. Increased serum levels of the chaperone Hsp70 have been identified in T2DM patients [95] and Hsp70 has been linked to inhibition of hIAPP aggregation [96]. Recent research using single-molecule two-colour coincidence detection suggested that Hsp70 inhibits aggregation by binding to heterogenic oligomeric intermediates of the protein, thus decreasing hIAPP toxicity [97]. Hsp70 obtained from the herb *Medicago sativa* administered to people with insulin resistance has shown promising results [98].

ER stress triggers the activation of signalling pathways, which lead to a process known as the unfolded protein response (UPR) in an attempt to restore ER homeostasis. Mechanisms of UPR include the increase in ER size, decrease in protein synthesis, increase in protein folding capacity and an increase in the removal of protein aggregates. If the UPR is unable to cope with increased amounts of aggregates forming in conditions such as T2DM, the accumulation of aggregates in the cells activates apoptotic pathways, leading to cell death and a decrease in β cell mass [79].

#### 2.6.4. Heparan Sulphate Proteoglycans

Heparan sulphate proteoglycans (HSPGs) vary in subcellular localisation and can be divided into one of four HSPG subtypes, depending on their location (at the cell surface, cell membrane, at the luminal site of intracellular vesicles or secreted). The latter have been identified as having an essential modulatory function in multiple developmental signalling processes through the regulation of molecules, including WNT transforming growth factor and fibroblast growth factor [99]. A type of secretory HSPG called perlecan has been observed binding to hIAPP through heparan sulphate glycosaminoglycan chains. This binding has been linked to an increase in hIAPP aggregation in vitro. In vivo experiments using transgenic hIAPP mice, (one group carrying a gene preventing perlecan binding, the other with wild type perlecan) showed a significant decrease in amyloid deposition in comparison to wild type after being fed a high fat diet for a year [100]. These results suggest a future potential therapeutic intervention via HSPG molecules such as perlecan.

#### 2.6.5. The Immune Response in Amyloidosis

Interleukin-1β (IL-1β) is an inflammatory cytokine that plays a role in immune responses to injuries and infections [101] and is a factor in T2DM pathogenesis. Increased amyloid formation has been linked to increased levels of IL-1β. Historic T2DM studies used IL-1β in ng/mL concentrations; however, recent papers have suggested that pg/mL concentrations represent more physiological in vivo β-cell islet concentrations [102]. When transgenic hIAPP mice islet cells were cultured in vitro with pg/mL concentrations of IL-1β, increased levels of IAPP were released, accelerating the formation of aggregates as well as decreasing β-cells in islet cultures, attributed to the cytotoxic effect of hIAPP on β-cells [103]. The heightened inflammatory response caused by increased levels of IL-1β contributes to a reduction in insulin secretion. In vitro experiments using interleukin 1 receptor antagonist (IL-1Ra) with hIAPP transgenic mice showed reduced levels of hIAPP amyloid, further suggesting that decreased IL-1 levels reduce formation of amyloid fibrils. Studies on mice and humans point towards M1 macrophages being present in increased numbers in obese individuals [104]. Obese individuals have adipose tissue containing up to 50% of macrophages present in the stromal vascular region, whilst non-obese individuals have less than 10%. There are several mechanisms proposed to account for the increased prevalence of macrophages in adipose tissue in obese individuals including increased secretion of chemokines, death of adipocytes, fatty acid flux dysregulation and hypoxia [105]. For many years, it has been reported that T2DM and obesity are linked, and the mild, but chronic, inflammatory state in obese individuals suggests another accelerating factor for hIAPP aggregation.

#### 2.6.6. Cross-Seeding of hIAPP and Other Amyloidogenic Proteins

Proteins involved in other protein misfolding diseases such as Aβ in Alzheimer’s disease have been observed directly interacting with hIAPP. Monomers are thought to interact via their β-sheet structures (found in both Aβ and hIAPP), leading to cross-seeding and the formation of hybrid fibrils in the brain [106]. In vitro Thioflavin T (ThT) assays revealed a slower lag phase and a faster growth phase in these hybrid fibril formations [107]. A cross-seeding study by Zhang et al. [108], involving hIAPP and Aβ in the presence of cell membranes, showed that the hybrid fibrils interacted with membranes and possessed morphology similar to that of hIAPP fibrils. Interaction with the cell membrane lipid bilayer caused a decrease in bilayer fluidity. This suggests that the toxicity of the hybrid fibrils may be similar to that of hIAPP-only fibrils [109]. The specific mechanism behind hIAPP and Aβ interactions is an important and active area of research. In addition to similarities in primary sequence, recent cryo-EM structures revealed a structural similarity between the backbones of a polymorph of amidated hIAPP (polymorph 1) and Aß 1–42 fibrils (containing S-shaped folds), suggesting these regions as important for cross-seeding one another at the fibril ends [42].

hIAPP aggregates have been observed in organs such as the brain, but it is unknown whether hIAPP is produced in those areas or it migrates there via the peripheral circulation [108]. Recent studies proposed that the overproduction of hIAPP aggregates is regulated by hIAPP being secreted via exosomes as a type of “detoxifying mechanism”. The exosomes can then be taken up by other cells such as neuronal cells. This would provide further evidence linking T2DM and Alzheimer’s pathogenesis [71].

Accumulating evidence suggests a link between Parkinson’s Disease (PD) and T2DM through protein interactions between alpha-synuclein (known to misfold in PD) and hIAPP. Co-aggregation of the two proteins has been linked to an increased risk of PD in T2DM patients [109]. Hybrid fibrils arise from both proteins cross-seeding with each other. In vivo studies demonstrated that alpha-synuclein injected into transgenic hIAPP mice resulted in accelerated hIAPP amyloid formation in β-cells [110]. These findings show that other amyloidogenic proteins with similar morphological structures can cross-seed with hIAPP and increase the rate of aggregation, as well as contribute to the pathology of not only T2DM, but other diseases associated with each protein.

#### 2.6.7. Zinc Ion Concentration

The concentration of zinc in the body is thought to contribute to T2DM. Zinc concentrations of 10–25 µM, which is the range found extracellularly, can decelerate the aggregation process by elongating the lag phase and decreasing the elongation rate of the fibrils, as shown in vitro, whilst the opposite is true for higher zinc concentrations [111]. Perhaps higher concentrations of zinc found in the cell granules assist in the prevention of hIAPP misfolding and cell surface damage by delaying aggregation and allowing controlled release to the bloodstream, preventing aggregation near the cell surface [111]. Insulin has been observed to be an inhibitor of hIAPP aggregation [112], and this inhibition may be dependent on the concentration of Zn^2+^, which also controls the equilibrium between insulin monomers, dimers and hexamers. A decrease in Zn^2+^ ions shifts the equilibrium of insulin oligomers in the direction of zinc-free monomers and dimers, which bind to IAPP monomers more efficiently than insulin hexamers. This may explain the observed link between loss of function mutations in the ZnT8 zinc ion transporter expressed on secretory granules with a decrease in hIAPP aggregation [112].

### 2.7. Type 2 Diabetes Therapeutics

There are currently several treatments being commonly used to treat those with T2DM, mostly insulin and metformin. However, since this disorder is closely linked to obesity, preventative methods such as changes in diet and fitness are usually suggested as the best preventative method for people at high risk of T2DM (e.g., through genetic factors). Results from a randomised controlled trial using 5238 participants demonstrated that without additional standard medical treatment, diet change and increased activity does not necessarily reduce the risk of T2DM pathology; however, it does reduce the risk of T2DM in people with increased glucose impairment [113].

For those with T2DM, the most common drug prescribed for the management of T2DM is metformin, part of the biguanide class of medications. It is thought to operate by minimising glucose production by hepatic cells and suppresses cell inflammation via interaction with metabolic organs (such as the gastrointestinal tract), immune cells, as well as influencing the activity of the glucose-lowering hormone glucagon-like peptide 1 [20]. The most common side effects of metformin are nausea, abdominal pain and diarrhoea, and whilst they are most commonly mild, they have been observed in up to 50% of patients, which is a high statistic considering it is the most widely prescribed medicine [114].

The different protein conformations that IAPP can misfold into have been proposed as viable treatment targets, since the oligomeric and fibrillar forms have been linked with pathology [25]. Various non-phenolic natural inhibitors have been proposed as potential treatments for T2DM, since there is evidence that they can reduce/disrupt aggregation through various mechanisms. These include destabilisation of oligomers and the stabilisation of monomers, converting amyloid fibrils into non-toxic oligomers and monomers by interacting with the hydrophobic regions which form the β-sheet structures [115]. Resveratrol, a polyphenol found in red wine, has been shown to activate the sirtuin enzymes, specifically SIRT1 [116]. SIRT1 is a conserved transcription factor for many biological processes and has been observed to upregulate lysosomes which degrade Aβ protein via the lysosomal pathway, as observed by Western blotting and quantitative proteomics [117]. The addition of resveratrol to the hIAPP-INS1 cell line resulted in a significant decrease in amyloid deposits, together with increased insulin secretion, compared with untreated cells [118]. However, more studies are still needed to assess the potential as a treatment for T2DM, since there are concerns about its limited bioavailability and potential cytotoxicity [119].

Manganese-salen chloride derivatives EUK-8 and EUK-134 that have been shown to possess antioxidant properties can also inhibit hIAPP amyloidosis [120]. The antioxidant probucol shows promising results for targeting hIAPP by reducing oxidative stress by decreasing ROS levels [121]. Succinobucol, (a phenol ester) is a more metabolically stable derivative of probucol and also shows promise as a potential treatment for amyloidosis [122].

An alternative treatment option proposes utilisation of the hIAPP oligomers for active immunisation. Studies in vivo by Bram et al. [123] showed that treatment with oligomers leads to the production of antibodies that bind to the oligomers and neutralise their cytotoxicity. Transgenic mice that developed hIAPP antibodies showed lower levels of amyloid, in addition to increased insulin levels and decreased fasting blood glucose [123].

## 3. Discussion

T2DM will soon be a medical emergency as levels of obesity increase and our populations live longer [13]. Since accumulating evidence links the presence of misfolded hIAPP in (the majority of) T2DM patients with β-cell damage, it is clear that hIAPP is central to pathology [22]. Accumulating data suggests that residues 20–29 are essential for amyloid formation [67]. While the initiating factors triggering hIAPP misfolding and aggregation are not yet conclusive, several factors that link increased hIAPP amyloidosis with T2DM have been identified. Despite a plethora of hIAPP structures deposited in the protein data bank, structures of bone fide in vivo/ex vivo assemblies have not yet been fully characterised. In vitro structures and those seeded from ex vivo samples have been elucidated using methods such as TEM and CD [36] and have provided us with good insight into the potential structures of biologically relevant conformations, such as the secondary β-sheet conformations of the fibrils. The observation of hIAPP oligomers in sera of T2DM patients now requires further experiments to isolate and characterise these structures, since they are clearly biologically relevant and deserve a comprehensive investigation [70].

The observation of polymorphic structures with varying residue orientations also implies that there may be variations between the structures of different fibrils, which could affect the formation of fibrils as well as their cytotoxicity [28,65]. One question that keeps arising when discussing PMDs is whether the fibrils themselves are a consequence or a cause of the disease, and this is still unclear in T2DM. Interestingly, all forms of hIAPP (monomers, dimers, oligomers, fibrils) have been associated with some level of cell damage [63,67,72,75] through mechanisms such as interaction with the cell membrane and an increase in membrane fluidity [74]. What is also interesting is the fact that not all T2DM patients have hIAPP deposits in the pancreas [124] and hIAPP aggregates have been observed in asymptomatic healthy individuals. This could, in part, be attributed to the methods employed to detect the aggregates and highlights the need for better screening and detection methods for detecting misfolded proteins, especially smaller oligomers that are difficult to detect due to their low abundance.

Cytotoxicity of hIAPP is an area of important research that needs to be investigated further to determine the precise molecular events involved. An essential part of understanding hIAPP aggregation is knowing what accelerates the process, as this also assists in identifying viable treatment options.

There is additionally a need to develop alternative treatments with minimal side effects [114]. In this review we have summarised the role of hIAPP, its structure and factors that may accelerate its misfolding and aggregation, highlighting several potential targets for treatment that would assist in the decrease/inhibition of hIAPP aggregation—hIAPP concentration, combating cell stress, administering molecular chaperones, administering HSPGs, decreasing IL-1β levels, targeting co-aggregating proteins and targeting specific hIAPP structures. hIAPP levels above a threshold concentration were observed to accelerate aggregation [77,78]. It may be that abnormal increased concentrations of hIAPP lead to amyloidosis. Thus, one intervention could be developing treatments that maintain hIAPP levels below this threshold value [66]. Zinc ion concentrations have been shown to affect the insulin oligomer equilibrium with lower concentrations that shift the equilibrium to favour hIAPP binding [112]. Cell stress such as ER stress and oxidative stress have also been observed to accelerate aggregation [7,79,80] contributing to the build-up of hIAPP in a conformation that is more prone to aggregate [81]. The build-up of hIAPP as well as its cytotoxicity further contributes to the pre-existing cell stress, further accelerating the pathology. As ER stress and oxidative stress are promoted by elevated fatty acid levels [81] and chronic hyperglycemia [7,82], preventative measures could be taken in order to reduce the risk of cell stress, in particular adopting a healthy diet.

Part of the ER stress response involves molecular chaperones [93,95]. Reduced chaperone expression has been shown to increase hIAPP aggregation, as seen with reduced levels of Hsp70 [96]. Administering Hsp70 and/or other chaperones may offer a novel treatment option [98]. Administering HSPGs is another viable option, as they have been shown to accelerate aggregation when their concentrations are low [100]. Molecules involved in the immune system response such as IL-1β, which is elevated due to chronic hyperglycaemia, have also been linked to acceleration of hIAPP aggregation. Decreasing IL-1β levels through injection with IL-1Ra in transgenic animal models have shown decreased aggregation rates, proposing yet another treatment option [103].

Other amyloidogenic proteins can accelerate the misfolding of hIAPP through cross-seeding and co-aggregate with hIAPP. Targeting other proteins may provide a good therapeutic target for T2DM, and vice versa, which has already been suggested in the literature [125].

Additionally, as more biochemical and structural information becomes available regarding the different forms of hIAPP, specific parts of these structures can be targeted when developing therapeutics. These include molecules to stabilise monomeric hIAPP to prevent formation of misfolded hIAPP, or those that bind to misfolded forms to block binding of additional hIAPP molecules or target misfolded forms for removal. Treatments can also target molecules in pathways associated with accelerating the misfolding and aggregation of hIAPP, as summarised in Figure 3; thus, they can be targeted to decrease hIAPP levels indirectly. For these studies to be successful, we need better in vitro and in vivo models such as better animal models and better availability of human samples from T2DM patients.

## 4. Conclusions

T2DM is a condition that is increasing at an alarming rate worldwide. In this review, we have highlighted a plethora of studies that link the protein hIAPP with T2DM. There are several factors linked to T2DM that have been shown to accelerate hIAPP amyloid formation by directly or indirectly interacting with hIAPP oligomers, monomers, and fibrils under hyperglycaemic conditions. Hyperglycaemia causes an environment which increases hIAPP aggregation. HIAPP aggregates and oligomeric species have been shown to be crucial for β-cell apoptosis and for T2DM development. Precisely what initiates these reactions is unclear, in addition to the precise molecular mechanisms involved in these pathways; however, amyloidosis and interactions with membranes appear to be essential factors linking hIAPP to T2DM.The propensity to aggregate lies in the primary structure of hIAPP, as seen in studies comparing rat and human IAPP (Figure 1). Understanding exactly how hIAPP misfolds and overcomes regulatory mechanisms, enabling them to accumulate in different parts of the body (including the brain and sera), is not fully understood and requires further in vitro and in vivo experiments. We have identified several metabolic factors that influence protein aggregation including cell stress, hIAPP and molecular chaperone concentration, the immune response and concentration of other amyloidogenic proteins which accelerate amyloidosis. All of these factors, in addition to the different hIAPP structural conformations, are all potential therapeutic targets that could reduce or prevent oligomer and fibril formation to decrease β-cell cytotoxicity and associated T2DM pathology. Understanding them and the way they interconnect is another valuable clue needed for solving the T2DM riddle.

## Figures and Tables

**Figure 1 life-12-00583-f001:**
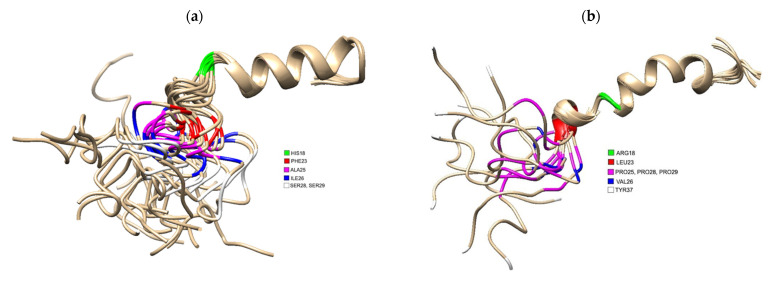
(**a**) 3D Structure of human IAPP residues 1–37 of the mature protein, solved by NMR (PDB 5MGQ) with key residues highlighted to show the difference compared to rat IAPP. The C and N terminus are located on the left and right, respectively. Residues were highlighted using USCF Chimera. HIS18 is highlighted in green, PHE23 is red, ALA25 is magenta, ILE26 is blue and SER28 and SER 29 are white.; (**b**) 3D Structure of rat IAPP residues 1–37, solved by NMR (PDB 2KJ7). Residues differing from hIAPP were highlighted using USCF Chimera. ARG18 is highlighted in green, LEU23 is red, PRO25, PRO28 and PRO 29 are magenta, VAL26 and TYR37 is white.

**Figure 2 life-12-00583-f002:**
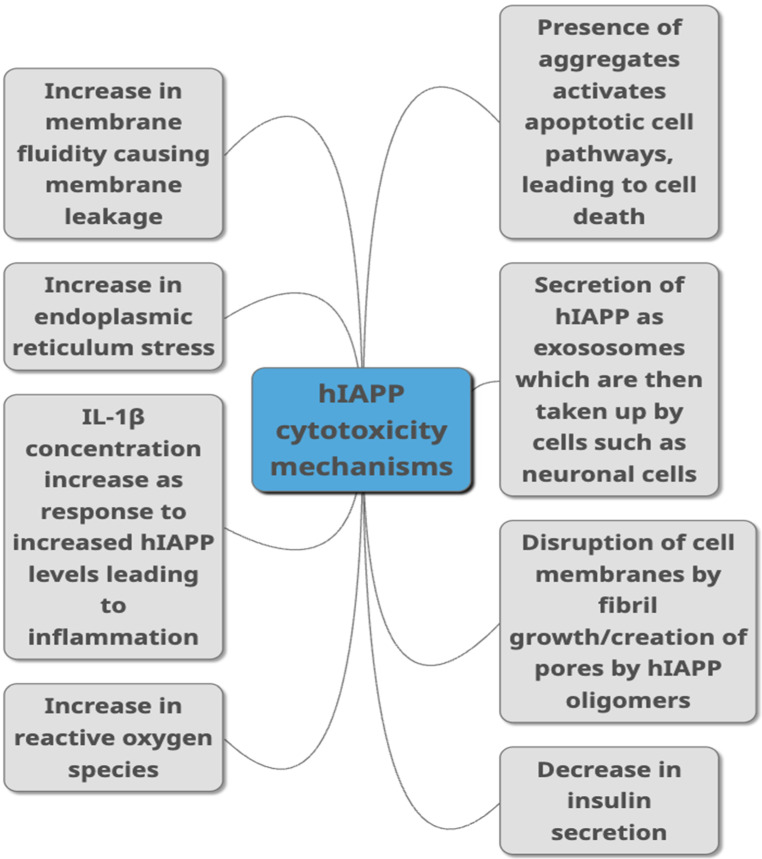
Proposed mechanisms through which hIAPP is linked to toxicity in T2DM as summarised from studies published in peer-reviewed journals. They include different ways of damaging the cells such as causing damage to the membrane, causing cell inflammation, damaging cells through aggregate build-up, by increasing ROS production and ER stress. Decreased insulin secretion, activation of apoptopic pathways and secretion via exosomes are other pathways suggested to link hIAPP with cytotoxicity.

**Figure 3 life-12-00583-f003:**
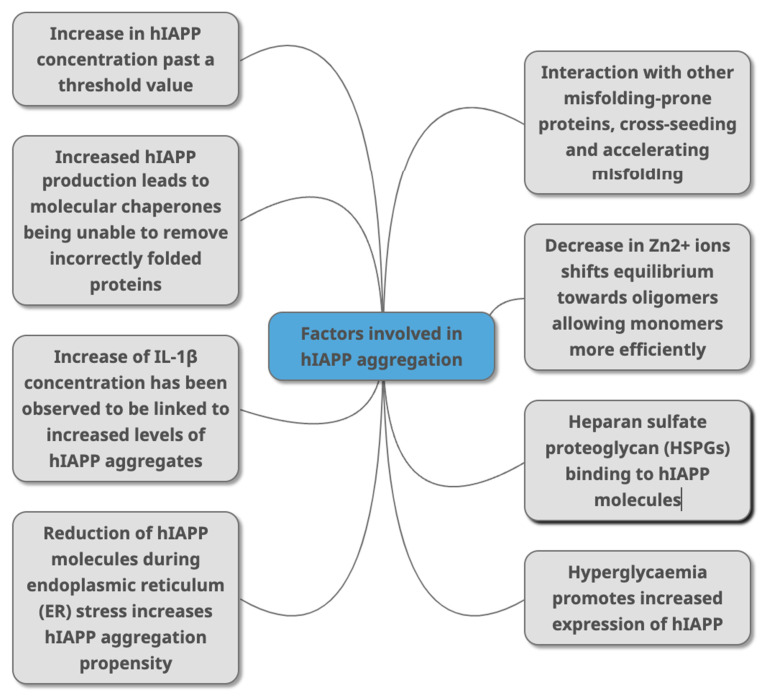
Factors that have been identified in peer -reviewed journals as contributing to hIAPP aggregation. This can occur due to change of concentrations of different molecules (HSPGs, Zn^2+^ IL-1β, Aβ, alpha synuclein) that directly or indirectly interact with hIAPP as part of various metabolic mechanisms (molecular chaperones, and through conditions linked with T2DM (hyperglycaemia, ER stress).

## Data Availability

Not applicable.

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
