# Peer review of "Factors That Contribute to hIAPP Amyloidosis in Type 2 Diabetes Mellitus"

_life, 2022, doi:10.3390/life12040583_

Round 1

Reviewer 1 Report

The paper extensively reviews what is known about hIAPP and its relationship to T2DM. It seems to cover all the most relevant points.

However, it fails to establish a connection between them and many times it "feels" like a simple list of the research carried out in the system. It lacks cohesion.

Repetitive. The same information is presented in the abstract, section 2.6, the discussion and the conclusions.

One of the most important features of IAPP is that it is an IDP. Which is not mentioned, and therefore the relationship with its function is not established.

The use of the term "conformation" is confusing. The IAPP is an IDP which means that it does not have a single conformation in solution, but many.

Other times it seems that the term refers to its oligomeric state.

Also what is the meaning of “mis-fold” for something without a unique fold (IDP)

When IAPP have different “conformations”? When does it interact with membranes? metals? etc

Paragraph line 42, Not just local information (UK), wordwide information would be good for a international paper.

Line 111: The information in this paragraph would be better understood if a figure with sequence is included.

Section 2.6 lists several molecules that can enhance fiber formation in vitro, but does not state the relationship of these molecules in vivo. can they ever be in touch?

line 222: it seems to refer to the core of the fiber, but in the previous paragraph, It was referring to the helix conformation.

paragraph: 227. oligomers. Just a single reference, are there no more oligomer studies?

Figure 1. White background would be better. Indicate the residues in the figure. The structure are the same, and it is a conformation when interacts with “membrans”.

Figure 3 and 4. The scheme is not very useful or visible. It has the same information that a table would have.

Include the references to search.

Author Response

The paper extensively reviews what is known about hIAPP and its relationship to T2DM. It seems to cover all the most relevant points. However, it fails to establish a connection between them and many times it "feels" like a simple list of the research carried out in the system. It lacks cohesion.

Thank you to reviewer 1 for taking the time to read our manuscript and providing us with useful feedback. We have now gone through each of your points and addressed them below.

Repetitive. The same information is presented in the abstract, section 2.6, the discussion and the conclusions.

Thank you for pointing this out.  The points we have raised in all of these sections are paramount to the paper. Whilst this may come across as repetitive, we feel that these points need to be repeated to reiterate the importance of these issues to the reader.

One of the most important features of IAPP is that it is an IDP. Which is not mentioned, and therefore the relationship with its function is not established.

Thank you to the reviewer for this comment. In the second paragraph of section 2.1 we had already written the text “hIAPP is a 37 amino acid protein with an element of intrinsically disordered structure”. Later on in section 2.2 we also reference the paper by Scollo et al 2020 (now reference 24) that refers to IAPP having a random coil structure and the fact IAPP is an amyloidogenic IDP. Since this point has been raised by the reviewer, we have decided to change the text slightly in section 2.1 as outlined below to highlight the fact that hIAPP is classified as an intrinsically disordered protein.

hIAPP is a 37 amino acid protein with an element of intrinsically disordered structure and is secreted from β-cells together with insulin [23]. It is classified as an intrinisically disordered protein (IDP), that lacks stable secondary or tertiary structures under physiological conditions, since it contains intrinsically disordered regions within its structure [24].

The use of the term "conformation" is confusing. The IAPP is an IDP which means that it does not have a single conformation in solution, but many. Other times it seems that the term refers to its oligomeric state. Also what is the meaning of “mis-fold” for something without a unique fold (IDP)

Thank you for reiterating this point and the fact that hIAPP is an IDP. From reading the literature, the precise structure of hIAPP and the different conformations that it can adopt during the process of amyloidosis and hence in disease (monomer, oligomer, fibril) is unclear. This is because all of the structures have been solved using hIAPP produced in vitro using a range of peptides and analysed using a range of structural techniques, hence this would explain the conflicting structures produced. This is why we have referred to the structures as ‘conformations’ in many places since the true structure in vivo is still not known. It is accepted in the field that hIAPP is a protein that can ‘misfold’ since it can form a different structure under certain conditions that links to disease, hence it is now widely accepted that type 2 diabetes is a protein misfolding disorder since hIAPP can form amyloid fibrils. We agree that the literature is confusing and have tried to provide an un-biased reviewed of the published literature to highlight the fact that hIAPP can adopt different ‘conformations’ yet the in vivo structure and the role that these different conformations play in type 2 diabetes remains unclear.

When IAPP have different “conformations”? When does it interact with membranes? metals? etc

Thank you for your comment. We have already written a section addressing the interaction with membranes that can be found in section 2.2 and a section dedicated to the importance of Zinc ions in type 2 diabetes can be found in section 2.6.7, both of which have also been copied below:

Since membranes have been suggested as crucial to hIAPP misfolding and aggregation [43] it is essential to characterise the structure of hIAPP when bound to a membrane-like environment. Studies suggest that membranes accelerate the transition from α-helical to β-sheet pleated structures as revealed by NMR, EPR, TEM and Atomic force microscopy (AFM) studies [44][45][46][47]. NMR and EPR experiments using hIAPP bound to micelles highlighted residues 9-22 as having an unstable α-helical region which is assumed to become stabilised during the aggregation process [32]. Lipid binding assays and TEM showed that when bound to anionic or zwitterionic membranes, hIAPP transitions from an α-helical structure to a β-sheet structure within minutes [48]. AFM has also been used to observe membrane-bound pre-fibrillar hIAPP, showing structures resembling ion-channels which could damage membranes by forming pores and changing their fluidity [49]. Other AFM studies have shown hIAPP accumulating on membranes in the form of unstructured aggregates. Membranes containing cholesterol revealed less bound amyloid compared to cholesterol-free membranes [50][51][52] highlighting the importance of membrane composition to amyloidosis.

2.6.7. Zinc ion Concentration

The concentration of zinc in the body is thought to contribute to T2DM. Zinc concentrations of 10-25 µM which is the range found extracellularly, can decelerate the aggregation process by elongating the lag phase and decreasing the elongation rate of the fibrils as shown in vitro, whilst the opposite is true for higher zinc concentrations [110].Perhaps higher concentrations of zinc found in the cell granules assist in prevention of hIAPP misfolding and cell surface damage by delaying aggregation and allowing controlled release to the bloodstream, preventing aggregation near the cell surface [110]. Insulin has been observed to be an inhibitor of hIAPP aggregation [111], and this inhibition may be dependent on the concentration of Zn2+ which also controls the equilibrium between insulin monomers, dimers and hexamers. A decrease in Zn2+ ions shifts the equilibrium of insulin oligomers in the direction of zinc-free monomers and dimers, which bind to IAPP monomers more efficiently than insulin hexamers. This may explain the observed link between loss of function mutations in the ZnT8 zinc ion transporter expressed on secretory granules with a decrease in hIAPP aggregation [111].

Paragraph line 42, Not just local information (UK), worldwide information would be good for an international paper.

Thank you for this suggestion. We agree, for a manuscript with an international readership, this information is not useful hence we have updated the text in the introduction section (section 1) and copied it below. We have also deleted the UK associated references and updated our reference list accordingly.

The World Health Organization conducted a worldwide study on diabetes in 2016  [8]. The report showed that the number of adults currently living with diabetes is over 422 million worldwide, that has quadrupled since 1980. T2DM accounts for 95% of all these cases resulting in over 1.5 million deaths. This increase has been linked to rises in obesity that has resulted in a 5% increase in mortality rate in those under the age of 70 with complications linked to blindness, stroke, heart attack, kidney failure and lower limb amputations.

Line 111: The information in this paragraph would be better understood if a figure with sequence is included.

Thank you to the reviewer for this suggestion. Whilst this is an interesting study, we feel that this does not warrant an individual figure since these structures were formed in vitro so we do not know whether this resembles what is happening in vivo. We do not want to draw attention to work that may later be disproved as further studies are conducted. In other protein misfolding disorders such as Alzheimer’s and Prion disease, it has already been shown that fibrillar forms produced in vitro differ in structure from those in vivo (Kollmer et al 2019  10.1038/s41467-019-12683-8, Terry et al 10.1038/s41598-018-36700-w) hence why we do not want to include this additional figure.

Section 2.6 lists several molecules that can enhance fiber formation in vitro but does not state the relationship of these molecules in vivo. can they ever be in touch?

Thank you to the reviewer for their comments on this. Whilst the precise mechanisms involved in fiber formation in vivo is not known, we feel that we have addressed the potential role of these molecules (and other factors) in fibre formation (and T2DM) by referring to the limited studies published.

For example in section 2.6.1. we have commented on the concentrations of hIAPP in secretory granules, in section 2.6.2 we refer to studies done with T2DM patients showing an increase in cell stress, and in section 2.6.3 we refer to studies showing increased Hsp70 levels in T2DM patients (known to be associated with hIAPP aggregation). In section 2.6.4., 2.6.5 and 2.6.6 we refer to in vivo rodent experiments that show perlecan, IL-1ß and other amyloidogenic proteins increase amyloid fibril formation respectively. As outlined in section 2.6.7, in vivo studies are urgently needed to assess the role of Zinc (and other metals) in fibril formation and T2DM.

line 222: it seems to refer to the core of the fiber, but in the previous paragraph, It was referring to the helix conformation.

Thank you for pointing this out. It is not yet clear what conformation(s) of hIAPP are associated with toxicity and exactly how they cause toxicity hence we have simply cited what has been reported in the literature, hence we wrote However, more than one form of hIAPP may contribute to cytotoxicity, possibly to different extents that may be via a variety of mechanisms’ to emphasise this uncertaintly.

paragraph: 227. oligomers. Just a single reference, are there no more oligomer studies?

It is unclear what region of the paper is being referred to here. There is no section 2.27 or text alongside line 227. If the comment is relating to oligomer studies, then yes there are only limited studies characterizing oligomers since they are thought to be short-lived intermediates present in low quantities hence, it is very difficult to study them in vitro and in vivo, hence the few citations.

Reviewer 2 Report

This is a nice review and provide a comprehensive summary on the factors that  contribute to hIAPP misfolding and aggregation. I recommended for publication  with the following comments.

1. (2.2) The structure of hIAPP

In addition to ref 34, there are several other recent reports on cryo-EM 
strcucture of hIAPP, which could be commented and cited. For example, (1) 
Röder, C., Kupreichyk, T., Gremer, L. et al. Cryo-EM structure of islet 
amyloid polypeptide fibrils reveals similarities with amyloid-β fibrils. Nat 
Struct Mol Biol 27, 660–667 (2020). (2) Gallardo, R., Iadanza, M.G., Xu, Y. 
et al. Fibril structures of diabetes-related amylin variants reveal a basis 
for surface-templated assembly. Nat Struct Mol Biol 27, 1048–1056 (2020).

2. Fig. 1

It would be better if the N- and C- termimus are labeled.

Fig. 1 legend, (a) residues 34-78, the numbering is not consistent with 1-37 as shown later. It could be re-numbered or (noted).

3. Line 217-221

There is a structure report on the S20G (Gallardo et al, NSMB 2020 as 
commented above). It could be commented here too.

4. Line 425-433

There are strcutre reports on cross-seeding, e.g., Röder et al, NSMB 2020, 
which could be commented here too.

5. Line 168

In my opinion, the in vitro fibrils seeded with extracts of patients would be 
best described as ex vivo, other than "resemble the structure of hIAPP in 
vivo."

Author Response

Reviewer 2

This is a nice review and provide a comprehensive summary on the factors that contribute to hIAPP misfolding and aggregation. I recommended for publication with the following comments.

  1. (2.2) The structure of hIAPP

In addition to ref 34, there are several other recent reports on cryo-EM structure of hIAPP, which could be commented and cited. For example, (1) Röder, C., Kupreichyk, T., Gremer, L. et al. Cryo-EM structure of islet amyloid polypeptide fibrils reveals similarities with amyloid-β fibrils. Nat 
Struct Mol Biol 27, 660–667 (2020). (2) Gallardo, R., Iadanza, M.G., Xu, Y. et al. Fibril structures of diabetes-related amylin variants reveal a basis for surface-templated assembly. Nat Struct Mol Biol 27, 1048–1056 (2020).

Thank you to reviewer 2 for reading our manuscript and for all of their useful feedback, all of which has been very helpful. We have addressed each of these points in detail below.

Thank you for mentioning these papers. We are already aware of these structures and have read these papers previously. We agree that we should have included these important structures as they provide more structural information regarding hIAPP. We have now updated the text to reflect these new additions (section 2.2) and included these papers in the reference list. In addition, we have also included additional cryo-EM papers listed below.

Additional papers added:

Cao Q, Boyer D, Sawaya M, Ge P, Eisenberg D. Cryo-EM structure and inhibitor design of human IAPP (amylin) fibrils. 2020.

Zhang X, Li D, Zhu X, Wang Y, Zhu P. Structural characterization and cryo-electron tomography analysis of human islet amyloid polypeptide suggest a synchronous process of the hIAPP1−37 amyloid fibrillation. Biochem Biophys Res Commun. 2020 Nov;533(1):125–31.

We have added the updated text to section 2.2:

Recently, several cryo-EM structures of recombinant hIAPP have been produced (Cao et al 2020; Gallardo et al 2020; Roder et al 2020; Zhang et al 2020) that show fibrils are composed of two protofilaments. The structures observed suggest that the process of fibrillisation may be synchronous (Zhang et al 2020) and the structural similarity of hIAPP fibrils to polymorphs of amyloid-β (Aβ) fibrils (Roder et al 2020) may explain how both fibrils can cross-seed one another.

  1. Fig. 1

It would be better if the N- and C- terminus are labeled.

Thank you for this advice. We agree, we should include this for clarity and have now updated the figure legend to reflect this. We have added the additional text “The C and N terminus are located on the left and right respectively”. To the figure legend.

Fig. 1 legend, (a) residues 34-78, the numbering is not consistent with 1-37 as shown later. It could be re-numbered or (noted).

Thank you for pointing out this inconsistency. We have now double checked this and can confirm that it should say 1-37. We have updated this in the figure legend.

  1. Line 217-221

There is a structure report on the S20G (Gallardo et al, NSMB 2020 as 
commented above). It could be commented here too.

Thank you for pointing this out. We are aware of these structures and perhaps removed this reference during editing. We will add this reference back in (plus reference to Cao et al 2020 and Roder et al 2020 who analysed the S20G fibrils) and refer to it in section 2.3.  We have added the updated text:

Cryo-EM structures reveal differences between wild type and S20G fibrils (Cao et al 2020; Gallardo et al 2020; Roder et al 2020) both showing two protofilaments and a left-handed twist but differences in features such as repeat and crossover lengths. Differences observed in the backbone conformations could explain how the early-onset T2D IAPP genetic polymorphism S20G can aggregate more readily than wildtype and may provide a structural explanation for surface-templated fibril assembly (Gallardo et al 2020).

  1. Line 425-433

There are structure reports on cross-seeding, e.g., Röder et al, NSMB 2020, 
which could be commented here too.

Thank you for bringing this to our attention. We have now referred to this paper in section 2.6.6 and updated the text. We have also updated our reference list to reflect this new addition.

We have added the updated text:

In addition to similarities in primary sequence, recent cryo-EM structures revealed structural similarity between the backbones of a polymorph of amidated hIAPP (polymorph 1) and Aß 1-42 fibrils (containing S -shaped folds) suggesting these regions as important for cross-seeding one another at the fibril ends (Roder at al 2020).

  1. Line 168

In my opinion, the in vitro fibrils seeded with extracts of patients would be 
best described as ex vivo, other than "resemble the structure of hIAPP in 
vivo."

Thank you to the reviewer for their opinion on this, this is an important point. We understand the reviewer’s point and after discussion, we have decided to keep the wording as it is. This is because it is only speculative that the fibrils seeded from islet cells from a T2DM donor resemble those in vivo since no ex vivo structure currently exists therefore, we cannot be 100% certain that these fibrils have the same morphology as disease-associated fibrils in vivo. Hence to avoid overinterpreting the data and making statements that could later be disproved, we would like to keep the text as it is using similar descriptions used by the authors who solved the structures (Cao et al).

Reviewer 3 Report

Review for the manuscript entitled “Factors that contribute to hIAPP amyloidosis in Type 2 Diabetes Mellitus”.

This manuscript fully discusses the relevance of amyloid to T2DM, including recent findings. It highlights the importance of amyloid as a target for T2DM treatment in the future.

Some minor comments are made.

1.(Sections 2.2/2.3/2.4) It might also be worth discussing that in hyperinsulinemia, wasting of the insulin-degrading enzyme can delay Amyloid metabolism.

2.(section 2.6) Mention could be made of anti-oxidant and anti-inflammatory therapies targeting Amyloid (e.g. superoxide dismutase and probucol).

3.(section 2.7) The fact that SIRT1 activated by resveratrol facilitates amyloid beta peptide degradation should also be mentioned.

Author Response

Reviewer 3

This manuscript fully discusses the relevance of amyloid to T2DM, including recent findings. It highlights the importance of amyloid as a target for T2DM treatment in the future.

Some minor comments are made.

1.(Sections 2.2/2.3/2.4) It might also be worth discussing that in hyperinsulinemia, wasting of the insulin-degrading enzyme can delay Amyloid metabolism.

We would like to thank reviewer 3 for taking the time to read our manuscript and for their very useful comments and feedback.

We agree, we should add additional text addressing IDE and hyperinsulinemia, this is a good suggestion. We have now added additional text to section 2.2 and references as outlined below.

Importantly, decreased insulin degrading enzyme (IDE) levels can lead to hyperinsulinemia which may contribute to insulin resistance [Pivovarova et al]. IAPP is a substrate for the insulin degrading enzyme (IDE) [Bennett et al 2000], a zinc metalloproteinase expressed in most cell types and also found in the circulation. Modulation of IDE activity could have consequences for amyloid formation by hIAPP and other amyloidogenic proteins [Sousa et al]. Inhibition of IDE increased amyloidosis and hIAPP cytotoxicity in a cell culture model [Bennett et al 2003], but not in cultured islets [Hogan et al]. In mice models acute administration of a potent IDE inhibitor demonstrated beneficial effects on glucose tolerance [Maianti et al] indicating potential for therapy of T2DM but given its wide distribution and other biochemical functions such as a chaperone and ubiquitin-proteosome pathway modulator, long term or untargeted use of inhibitors may be problematic [Bennett et al 2003; Tang et al].

Pivovarova, O.; Höhn, A.; Grune, T.; Pfeiffer, A.; Rudovich, N. Insulin-degrading enzyme: new therapeutic target for diabetes and Alzheimer’s disease? Annals of Medicine 2016, 48, 614-624.

Bennett RG, Duckworth WC, Hamel FG. Degradation of amylin by insulin-degrading enzyme. J Biol Chem. 2000 Nov 24;275(47):36621-5. doi: 10.1074/jbc.M006170200. PMID: 10973971.

Sousa L, Guarda M, Meneses MJ, Macedo MP, Vicente Miranda H. Insulin-degrading enzyme: an ally against metabolic and neurodegenerative diseases. J Pathol. 2021 Dec;255(4):346-361. doi: 10.1002/path.5777. Epub 2021 Sep 17. PMID: 34396529.

Bennett RG, Hamel FG, Duckworth WC. An insulin-degrading enzyme inhibitor decreases amylin degradation, increases amylin-induced cytotoxicity, and increases amyloid formation in insulinoma cell cultures. Diabetes. 2003 Sep;52(9):2315-20. doi: 10.2337/diabetes.52.9.2315. PMID: 12941771.

Hogan MF, Meier DT, Zraika S, Templin AT, Mellati M, Hull RL, Leissring MA, Kahn SE. Inhibition of Insulin-Degrading Enzyme Does Not Increase Islet Amyloid Deposition in Vitro. Endocrinology. 2016 Sep;157(9):3462-8. doi: 10.1210/en.2016-1410. Epub 2016 Jul 12. PMID: 27404391; PMCID: PMC5007890.

Maianti JP, McFedries A, Foda ZH, Kleiner RE, Du XQ, Leissring MA, Tang WJ, Charron MJ, Seeliger MA, Saghatelian A, Liu DR. Anti-diabetic activity of insulin-degrading enzyme inhibitors mediated by multiple hormones. Nature. 2014 Jul 3;511(7507):94-8. doi: 10.1038/nature13297. Epub 2014 May 21. PMID: 24847884; PMCID: PMC4142213.

Tang WJ. Targeting Insulin-Degrading Enzyme to Treat Type 2 Diabetes Mellitus. Trends Endocrinol Metab. 2016 Jan;27(1):24-34. doi: 10.1016/j.tem.2015.11.003. Epub 2015 Dec 2. PMID: 26651592; PMCID: PMC4698235.

2.(section 2.6) Mention could be made of anti-oxidant and anti-inflammatory therapies targeting Amyloid (e.g. superoxide dismutase and probucol).

Thank you, this is an important area of anti-amyloid therapeutics, and we agree we should include some additional text to discuss their use as an amyloid therapeutic.  

We have added the additional text to section 2.7 (not 2.6 as it seemed more appropriate to add it to the therapeutic section) plus cited additional references as outlined below.

Manganese-salen chloride derivatives EUK-8 and EUK-134 that have been shown to possess anti-oxidant properties have can also inhibit hIAPP amyloidosis (Bahramikia, S., et al., 2013). The antioxidant probucol shows promising results for targeting hIAPP by reducing oxidative stress by decreasing ROS levels (Gorogawa, S., et al., 2002). Succinobucol, (a phenol ester) is a more metabolically stable derivative of probucol and also shows promise as potentially treatment for amyloidosis (Lo, M., et al., 2013)

Bahramikia, S.; Yazdanparast, R. Inhibition of human islet amyloid polypeptide or amylin aggregation by two manganese-salen derivatives. European Journal of Pharmacology 2013, 707, 17-25.

Gorogawa, S.; Kajimoto, Y.; Umayahara, Y.; Kaneto, H.; Watada, H.; Kuroda, A.; Kawamori, D.; Yasuda, T.; Matsuhisa, M.; Yamasaki, Y.; Hori, M. Probucol preserves pancreatic β-cell function through reduction of oxidative stress in type 2 diabetes. Diabetes Research and Clinical Practice 2002, 57, 1-10.

Lo, M.; Lansang, M. Recent and Emerging Therapeutic Medications in Type 2 Diabetes Mellitus. American Journal of Therapeutics 2013, 20, 638-653.

3.(section 2.7) The fact that SIRT1 activated by resveratrol facilitates amyloid beta peptide degradation should also be mentioned.

Thank you to the reviewer for this suggestion. We initially did not include this study as it was more focused on amyloid beta (not hIAPP) however, after discussion, we agree it is worth mentioning these observations in our review. We have included the additional text in section 2.7 and references below to reflect this.

Resveratrol, a polyphenol found in red wine, has been shown to activate the sirtuin enzymes specifically SIRT1 (Borra et al 2005). SIRT1 is a conserved transcription factor for many biological processes and has been observed to upregulate lysosomes which degrade Aβ protein via the lysosomal pathway as observed by Western blotting and quantitative proteomics (Li, et al., 2018).

Li, M.; Ji, J.; Zheng, L.; Shen, J.; Li, X.; Zhang, Q.; Bai, X.; Wang, Q. SIRT1 facilitates amyloid beta peptide degradation by upregulating lysosome number in primary astrocytes. Neural Regeneration Research 2018, 13, 2005.

Borra MT, Smith BC, Denu JM. Mechanism of human SIRT1 activation by resveratrol. J Biol Chem. 2005 Apr 29;280(17):17187-95. doi: 10.1074/jbc.M501250200. Epub 2005 Mar 4. PMID: 15749705.

Round 2

Reviewer 1 Report

Many times repeating something is not stressing something, it is simply repeating it.

As far as I know, the conformation and the oligomeric state are not the same. A protein can have the same conformation while forming dimers or tetramers or another oligomeric state.

About Figure 1 and the sequence. What I meant is that it would be useful to show the amino acid sequence of the protein in figure 1. But now, looking at your answer, everything you wrote can be applied to the structures you show in figure 1 (structures in specific condition in vitro). Why would this apply to other figures (I have no idea which ones) but not this one?